# FoodOmicsGR_RI: A Consortium for Comprehensive Molecular Characterisation of Food Products

**DOI:** 10.3390/metabo11020074

**Published:** 2021-01-27

**Authors:** Georgios Theodoridis, Alexandros Pechlivanis, Nikolaos S. Thomaidis, Apostolos Spyros, Constantinos A. Georgiou, Triantafyllos Albanis, Ioannis Skoufos, Stavros Kalogiannis, George Th. Tsangaris, Athanasios S. Stasinakis, Ioannis Konstantinou, Alexander Triantafyllidis, Konstantinos Gkagkavouzis, Anastasia S. Kritikou, Marilena E. Dasenaki, Helen Gika, Christina Virgiliou, Dritan Kodra, Nikolaos Nenadis, Ioannis Sampsonidis, Georgios Arsenos, Maria Halabalaki, Emmanuel Mikros

**Affiliations:** 1Laboratory of Analytical Chemistry, Department of Chemistry, Aristotle University of Thessaloniki, 54124 Thessaloniki, Greece; al_pechliv@hotmail.com (A.P.); cr_virgi@hotmail.com (C.V.); drity.kodra@gmail.com (D.K.); 2Biomic_Auth, Bioanalysis and Omics Laboratory, Center for Interdisciplinary Research and Innovation (CIRI-AUTH), Balkan Center, B1.4, 10th Km Thessaloniki-Thermi Rd, P.O. Box 8318, 57001 Thessaloniki, Greece; atriant@bio.auth.gr (A.T.); gagavou@bio.auth.gr (K.G.); 3Laboratory of Analytical Chemistry, Department of Chemistry, National and Kapodistrian University of Athens, Panepistimioupolis, Zografou, 15771 Athens, Greece; ntho@chem.uoa.gr (N.S.T.); ankritik@chem.uoa.gr (A.S.K.); mdasenaki@chem.uoa.gr (M.E.D.); 4Department of Chemistry, University of Crete, Voutes Campus, 71003 Heraklion, Greece; aspyros@uoc.gr; 5Chemistry Laboratory, Department of Food Science and Human Nutrition, Agricultural University of Athens, 75 Iera Odos, 11855 Athens, Greece; cag@aua.gr; 6Department of Chemistry, University of Ioannina, 45110 Ioannina, Greece; talbanis@uoi.gr (T.A.); iokonst@uoi.gr (I.K.); 7Laboratory of Animal Health, Food Hygiene and Quality, Department of Agriculture, University of Ioannina, 47100 Arta, Greece; jskoufos@uoi.gr; 8Department of Nutritional Sciences & Dietetics, International Hellenic University, Sindos Campus, 57400 Thessaloniki, Greece; kalogian@nutr.teithe.gr (S.K.); isampsonides@gmail.com (I.S.); 9Proteomics Research Unit, Biomedical Research Foundation of the Academy of Athens, 11527 Athens, Greece; gthtsangaris@bioacademy.gr; 10Department of Environment, University of the Aegean, 81100 Mytilene, Greece; astas@env.aegean.gr; 11Department of Genetics, Development and Molecular Biology, Aristotle University of Thessaloniki, 54124 Thessaloniki, Greece; 12Department of Medicine, Laboratory of Forensic Medicine & Toxicology, Aristotle University of Thessaloniki, 54124 Thessaloniki, Greece; gkikae@auth.gr; 13Laboratory of Food Chemistry and Technology, School of Chemistry, Aristotle University of Thessaloniki, 54124 Thessaloniki, Greece; niknen@chem.auth.gr; 14Department of Veterinary Medicine, School of Health Sciences, Aristotle University of Thessaloniki, 54124 Thessaloniki, Greece; arsenosg@vet.auth.gr; 15Department of Pharmacy, National and Kapodistrian University of Athens, Panepistimioupoli Zografou, 15771 Athens, Greece; mariahal@pharm.uoa.gr (M.H.); mikros@pharm.uoa.gr (E.M.)

**Keywords:** metabolomics, genomics, authenticity, traceability, nutritional value, food composition

## Abstract

The national infrastructure FoodOmicsGR_RI coordinates research efforts from eight Greek Universities and Research Centers in a network aiming to support research and development (R&D) in the agri-food sector. The goals of FoodOmicsGR_RI are the comprehensive in-depth characterization of foods using cutting-edge omics technologies and the support of dietary/nutrition studies. The network combines strong omics expertise with expert field/application scientists (food/nutrition sciences, plant protection/plant growth, animal husbandry, apiculture and 10 other fields). Human resources involve more than 60 staff scientists and more than 30 recruits. State-of-the-art technologies and instrumentation is available for the comprehensive mapping of the food composition and available genetic resources, the assessment of the distinct value of foods, and the effect of nutritional intervention on the metabolic profile of biological samples of consumers and animal models. The consortium has the know-how and expertise that covers the breadth of the Greek agri-food sector. Metabolomics teams have developed and implemented a variety of methods for profiling and quantitative analysis. The implementation plan includes the following research axes: development of a detailed database of Greek food constituents; exploitation of “omics” technologies to assess domestic agricultural biodiversity aiding authenticity-traceability control/certification of geographical/genetic origin; highlighting unique characteristics of Greek products with an emphasis on quality, sustainability and food safety; assessment of diet’s effect on health and well-being; creating added value from agri-food waste. FoodOmicsGR_RI develops new tools to evaluate the nutritional value of Greek foods, study the role of traditional foods and Greek functional foods in the prevention of chronic diseases and support health claims of Greek traditional products. FoodOmicsGR_RI provides access to state-of-the-art facilities, unique, well-characterised sample sets, obtained from precision/experimental farming/breeding (milk, honey, meat, olive oil and so forth) along with more than 20 complementary scientific disciplines. FoodOmicsGR_RI is open for collaboration with national and international stakeholders.

## 1. Introduction

In recent years, there has been an increased awareness towards the evaluation of food quality and safety in relation to human health and life. Existing legislation refers to technological processes of food, safety assessment by monitoring food contaminants, characterization of food composition and the determination of characteristic food nutrients. The “classical” targeted analytical procedures are still dominant; however, the holistic characterization of food products through “omics” technologies is gaining interest and applicability. Such novel approaches can provide new indicators of food quality, i.e., geographic and/or genetic origin, processing methods, adulteration, organoleptic characteristics, freshness, and so forth. Omic technologies are increasingly employed in nutrition studies aiming to increase the marketability of food products and to demonstrate and/or monitor their beneficial effects on human health. Nutrigenomics, Nutrigenetics and Nutritional metabolomics are topics and terms that are increasingly found in the literature and the wider public media.

Foodomics [1] combines food/nutrition sciences with advanced analytical techniques and bioinformatics, applying a hypothesis-free approach to globally map the composition of foods or biological fluids of food consumers, to elucidate critical questions, and address new challenges of a globalized world. These may include: safety/security, sustainability, authentication, traceability, improvement of food production in response to environmental changes, the potential use of food waste and other fields. This holistic perspective requires investments in research expertise and instrumental facilities, but in return it provides big data, which after the application of advanced statistical analysis and bioinformatics can generate new knowledge, highlighting unlooked-for patterns and associations of biomolecules. The latter can lead to the evidence-based development of new products or services. The potential application of foodomics is very wide. An indicative list includes the following: 1. comparison of foods composition; 2. enhanced control for food traceability and authenticity; 3. effective monitoring of maturation/ageing or storage or various technological processes applied in food production; 4. monitoring of freshness; 5. linkage of organoleptic properties with the molecular content; 6. scrutiny of health claims of functional foods; 7. revealing underlying trends and discriminatory patterns induced by food ingredient(s) at different expression levels; 8. studies of food safety. Overall, foodomics can provide new knowledge and promote the understanding of biochemical, molecular and cellular mechanisms that trigger advantageous or adverse effects of food components [2].

Foodomics is applied in various R&D aspects. In the study of the origin and quality of foods, foodomics can reveal discriminatory markers and, most importantly, patterns that characterize foods according to their geographical origin or subtle pedoclimatic effects (e.g., the altitude of production), the different legitimate or non-legitimate regimens/interventions applied by food producers (e.g., pharmaceutical-steroid-hormone treatment, feeding patterns for animal or insects, use of fertilisers in crop production, etc.). Such data are important to support marketability and branding, especially for Protected Designation of Origin (PDO) and Protected Geographical Indication (PGI) products.

In food safety, foodomics research can assist in the discovery of (bio)markers and mechanisms of toxicity and facilitate the identification of potential hazards. Untargeted-holistic approaches may reveal underlying and unlooked-for mechanisms of toxicity, such as polymorphisms or sub-toxic (but increased and unwanted) concentrations of biomolecules. It can be a potential tool for the investigation of food allergies and the analysis of transgenic foods, providing further insight on the effect of unintended transformations [2,3].

In nutrition studies, foodomics can provide new knowledge on the effect of nutrition and the fate of nutrients in the human body; Foodomics can identify (among others) perturbations caused by certain dietary regimens and characteristic patterns that can allow the development of a personalized dietary intervention for certain groups (e.g., athletes, pregnant women), or patients with food-related disorders [4].

Considering the needs for R&D in these topics, the Greek Secretariat for Research and Technology announced a call for new Research Infrastructures. To address the call, a consortium was formed and submitted a successful proposal that provided the means for the establishment of FoodOmicsGR_RI. FoodOmicsGR_RI is a distributed multidisciplinary national research infrastructure aiming to perform, assist and promote omics research in agri-food R&D fields in the Greek and international research environment. This research may have a strong impact on the Greek agri-food sector. Greece has a unique landscape at the crossroads of Europe with Asia and Africa. The land expands to distances of up to 1200 km in a mostly mountainous mainland. The country has a very lengthy coastline (13,676 km) and close to 3000 islands. Varying types of soil (including several volcanic types), water supply and pedoclimatic conditions result in a wide array of ecosystems and landscapes that give rise to one of the highest biodiversities in Europe, especially taking into account the area of the country. Flora includes 5752 species (452 endemic subspecies); the fauna includes 23,130 animal species, of which 3956 are characterised as Greek endemics. Furthermore, 3500 marine animal species and a high number of fungi species have been recorded in the Greek marine environment. Besides, a high proportion of these species are unique worldwide and this biodiversity results in the unique composition of Greek agricultural products.

Greece has been an agricultural country, where until the 1960s the majority of the population lived in rural areas as farmers, maintaining traditional habits in life, farming and cuisine. All these factors converge to create a very large collection and a very strong portfolio of unique and characteristic food products. Highlighting these characteristics through advanced technology offers a way to support quality, nutritional or health claims, increase their attractiveness and marketability, and provide added value to local products. The central scope of FoodOmicsGR_RI is to serve the needs of the agri-food producers, providing concrete, solid and comprehensive data on the composition and nutritional value of their produce.

## 2. FoodOmicsGR_RI, Structure

FoodOmicsGR_RI is a multidisciplinary consortium that joins forces from eight Greek Universities and Research Centers. FoodOmicsGR_RI brings together BIOMIC, a new interdisciplinary analytical laboratory of the Aristotle University of Thessaloniki (http://biomic.web.auth.gr/) with specialist groups from the same University, and groups of researchers from the Universities of Athens, Crete, the Aegean, Ioannina, the Agricultural University of Athens, the International Hellenic University and the Biomedical Research Center of the Academy of Athens. This consortium is highly complementary, comprising scientists from 17 Departments of the participating Institutes (a full list is given in the Appendix A (Structure of RI), and on our webpage http://foodomics.gr/).

FoodOmicsGR_RI core (bio)chemical analysis groups provide expertise in bioanalysis, food analysis, metabolomics, elemental metabolomics, genomics and proteomics. Central technologies include analytical separations, Mass Spectrometry (MS, both organic and elemental MS), Nuclear Magnetic Resonance (NMR) spectroscopy and Inductively Coupled Plasma Spectrometry. Analytical groups collaborate with (bio)informatics groups, food (bio)chemistry groups and application-oriented researchers with expertise in fields relevant to food and nutritional sciences: animal husbandry, veterinary pathology, fishery, apiculture, plant physiology, pomology, horticulture, vine and viticulture, environmental control, natural products, nutrition biochemistry and human exercise biochemistry. The core analytical groups have the critical mass to realise cutting-edge research in their specific field. Investment in infrastructure (Section 3.2) and the Analytical Portfolio (Section 3.3) promotes the analytical arsenal of these core groups, enabling them to provide higher quality research services to end-users from academia, government and industry. Despite a decade of economic crisis and brain drain, the FoomicsGR_RI partnering laboratories have managed to increase their personnel, instrumental capabilities, sophistication of expertise, portfolio of methods and applications to different studies and research hypotheses. Apart from the analytical capacities, FoodOmicsGR_RI members have gathered experience and data from the analysis of several thousands of samples of different types (each group with their own technology) either in untargeted modes or in a fully targeted mode, reporting on solid quantitative data in compliance with validation guidelines. It is important to mention that the long-lasting collaborations with a large number of researchers from the private sector (food producers, clinics, CRO laboratories, life science companies and analytical instrument/technology vendors), as well as international research organisations and different regulatory bodies and authorities (Eurachem, NIST, mQACC, Norman network, IRMM JRC and other national and international bodies).

### 2.1. Governance

As a major initiative, the governance of FoodOmicsGR_RI is a demanding task that expands at various levels. It includes the General Assembly (GA) that is composed of delegates from all involved institutions and a Steering Committee of five members from the participating institutions that manage the operation of the RI. An International Expert Advisory Board (IEAB) has been appointed with the scope to further enrich the knowledge, the experience and the network of the RI participants. The 11 IEAB members include scientists from consumer organisations, industry or other businesses, potential stakeholders, and representatives of the activities of the RI.

### 2.2. Access to Facility

The facility is open for collaboration subject to the conditions detailed at the central FoodOmicsGR_RI web page with templates for access-use of infrastructure, usage policy, protocols for sampling, sample delivery, and information about the different FoodOmicsGR_RI SOPs. The services offered by FoodOmicsGR_RI can be tailor-made including all or combinations of the following:-Consultancy services and uptake of experimental intervention for foodomics studies (food product -oriented studies) or nutritional studies.-Big Data Handling and statistical analysis by various modes and special software including bespoke scripts and algorithms.-Genomic, proteomic and metabolomic analysis of foods, food products and biological samples from nutritional studies.-Quantitative Determination of key target molecules (nutrients, contaminants or other) in foods and biological samples.

FoodOmicsGR_RI can provide support for the realization of studies on small molecules (metabolomics), and cover genetics/genomics/epigenetics and proteomic fields, offering a “one stop-shop” model to stakeholders and users.

A detailed list of validated protocols is given in Appendix A. The combination of strong expertise, strong links with instrument manufacturers, software developers and analytical material companies provides the base for the establishment of state-of-the-art methods and technologies. As a result, the facility is uniquely placed to realise cutting edge research.

## 3. FoodOmicsGR_RI Plan and Strategy

The ideas and scope of FoodOmicsGR_RI are aligned with several of Research and Innovation Strategies for Smart Specialisation (RIS3) priorities for the agri-food sector. A list of relevant priorities is given in Appendix A, which shows priorities related to food characterization and the development of new products, and Appendix A, which shows priorities that are relevant to nutrition, health and wellness. The main aim is to provide a focal point for the realization and initiation of R&D efforts from research institutes, universities and the private sector to address such priority topics. A further aim is to provide the basis for synergistic efforts and ideas that will generate new knowledge and new research aims. The facility plans to integrate research and development (R&D) and services, and play a central role in the realization of R&D activities that can be grouped in two main directions: (1) providing data from analysis of food and agricultural products to gain new knowledge and enhance their position in the international market (Section 3.5) and (2) to investigate the relationship between nutrition, health and wellness (Section 3.6).

Omic approaches can become a tool to broaden the knowledge and understanding of how biodiversity (genotypic variation) can improve food and nutrition, influencing national policies and markets. This can be achieved via the examination of other nutrient-rich crops and plant species, not yet fully utilized, or unexplored traditional cultivars (e.g., olive tree cultivars, grapevine cultivars), benefiting both producers and consumers.

Implementation: The strategy to approach such multidisciplinary and multifaceted work is to overcome the fragmentation of research by organising paths for effective samples’ analyses and, similarly, paths for data curation that combine omics approaches to provide holistic analytical results and highlight characteristic features. The outline of specific sub-projects is described in the following sections.

### 3.1. Literature Analysis of the Composition of Greek Foods and Compilation of Greek Food Composition Databases

Foodomics research requires state of the art technologies and methods but also the existence of specialists’ knowledge and high-quality sample banks, which, after comprehensive analysis, can provide data for the establishment of compositional databases. Compositional databases are the best means to provide the basis for the control of real samples (food products). Our consortium started this work with an exhaustive literature review. This resulted in the publication of topical reviews on metabolomics of different foods, such as the olive drupe and olive oil [5,6,7], grape and marc spirits [8], fruit juices [9] and wine [10]. Consortium members have also contributed to the documentation of the full body of research work dealing with NMR-based food analysis and metabolomics by publishing a monograph on the subject [11], and covering subsequent work in the field via extensive literature reviews [12,13].

Additionally, Special Issues organised by FoodOmicsGR_RI members in 2020 include the following:

“Metabolomics in Food Authentication: Strategies and Applications” in the MDPI Molecules Journal by Prof. Thomaidis and Dr. Dasenaki (guest editors), including important foodomics-based authenticity studies.

“Nutritional Metabolomics” in the MDPI Metabolites Journal, guest-edited by Prof. Georgios Theodoridis, Dr. Christina Virgiliou and Dr. Olga Deda.

“Metabolomics Methodologies and Applications II”, guest-edited by professor Helen Gika, in the MDPI Metabolites Journal.

Following this fundamental work, the consortium works further to generate databases of the molecular and elemental composition of foods to facilitate control, stir further technological developments and discover relationships between food constituents and/or elemental signatures with properties sought by consumers and producers. The generation of such databases started by thorough data-mining and text-mining of the literature and collaboration with expert groups from the control sector. As a starting step, nine foods/food products have been studied as priority commodities: olive oil, olives, wine, milk, cheese, yogurt, honey, royal jelly and pollen. Since the scope is to generate a database of the contents of Greek foods, a filter is set to study only publications that report Greek products, and exclude publications that map products and varieties that are not found or those that do not originate from Greece. Emphasis was given to map specific varieties/types of Greek produce. To generate data for a certain food, a set of keywords was selected; i.e., for wine, this includes characteristic Greek variety names: Asyrtiko, Nemea, Xinomauro and so forth.

Text mining algorithms developed in our group are used to mine the international literature for publications containing carefully selected keywords of interest. The codes seek to gather quantitative information reporting the concentration of food constituents such as elements and small molecules. This information is subsequently curated by PhD level scientists to verify its validity and relevance, with extra caution given at concentration numbers and units.

The generated database is now formatted as a web-based database to be soon open to the research community. During the realization of the works of FoodOmicsGR_RI, new data will be generated from FoodOmicsGR_RI sub-projects. Sample sets are analysed by various information-rich analytical platforms. The data are fused to generate an atlas of the content of these unique samples. New analytical data will populate, control, corroborate the data in the database. This strategy improves confidence in findings, while it can also reduce the impact of confounding factors (effect of climate, disease, farming techniques and so on) in large-scale analysis and help researchers to reach useful, statistically meaningful findings. The overall aim of this activity is to establish a curated, internet-based database hosted in FoodOmics portal, that would serve as a reference point for the search of the content (providing concentration values) of Greek food products.

Regarding milk, cheese and animal tissue samples, our expert research teams have also built a reference database through the implementation of various research projects [14,15,16,17,18,19,20,21,22,23,24].

### 3.2. Advancing Analytical Capability of the RI

The central aim of FoodOmicsGR_RI is to strengthen the existing infrastructure of its partners and facilitate basic and applied translational research. With this upgrade, the aggregate capability for capital instrumentation of the laboratories reaches five hybrid time of flight mass spectrometers Q-TOF-MS hyphenated to UPLC systems (two with ion mobility capability(TIMS-TOF-MS)), a UPLC-QTRAP 6500+ MS/MS, more than 10 triple quadrupole MS/MS instruments hyphenated to UPLC, four UPLC-Orbitrap HRMS (one with nano-LC), three HPLC-MS (High-performance liquid chromatography mass spectrometry) single quadrupole instruments, an SFC-UV/MS system, more than 8 GC-MS (Single quadrupole analysers), and three GC-MS/MS (Gas chromatography–mass spectrometry systems with triple quadrupole MS). Mass Spectrometry capabilities of FoodOmicsGR_RI include Matrix-assisted laser desorption/ionization- time of flight- mass spectrometry (MALDI-TOF MS) exploiting the potential for high molecular weight analysis, enhancing the development of rapid and reliable high-throughput methodologies for food authenticity studies. NMR spectroscopy capabilities include two 600 MHz, two 500 MHz and four 400 MHz NMR spectrometers. Elemental analysis is performed in two ICP-MS Inductively coupled plasma- mass spectrometry) and two (Inductively coupled plasma - optical emission spectrometry) ICP-OES instruments. In addition, virtually all necessary instrumentation is available in the laboratories of the partners including FT-IR (Fourier-transform infrared spectroscopy), NIR (near infrared spectroscopy), capillary electrophoresis (CE)-, and several HPLC and GC instruments linked with various detectors: UV, PDA (Photodiode Array Detector), fluorescence, ELSD (Evaporative light scattering detector) for LC; FID (flame ionization detector), ECD (Electron capture detector) for GC). FoodOmicsGR_RI involves strong informatics expertise and access to high computing power. Staff scientists and newly recruited researchers are working on data treatment and the implementation of fusion of data collected from the different analyses and omics platforms.

### 3.3. Advancing and Harmonising the Analytical Portfolio

Advancing and harmonising the available analytical methods is a central task. The RI partners have very strong track records in research, as evidenced by their publications/presentations and patents awarded. Partner selection provides high complementarity in methods, expertise, infrastructure and application orientation. The latter practically covers the whole Greek agri-food sector. An array of protocols for genetic/metabolomics/proteomics analysis in targeted and untargeted mode exists among the FoodOmicsGR_RI partners and a list of current protocols and methods is given in Appendix A; this list will be frequently updated, so the reader may visit the webpage for updated information. New methods are currently developed and validated. Further to this, our core analytical separation groups design collaborative work to map the metabolome, applying complementary separation modes, e.g., by using HILIC and RPLC separation modes.

FoodOmicsGR_RI has established a multitude of methods for metabolic profiling as well as elemental, genetic and proteomic profiling. In addition, protocols exist for the analysis of nutrients and pollutants in foods and the characterization of foods by state-of-the-art technologies. These cutting-edge analytical tools are applied in the quest to reach a thorough characterization of food products with a focus on Greek products of high value and market potential, which includes but is not limited to olive oil, wine, honey, dairy products and seafood. Figure 1 provides an outlook of the major technologies and applications of the facility.

#### 3.3.1. MS Based-Metabolomics

A major concentration of analytical force is directed to the small molecule domain. Metabolic profiling or metabolomics is the largest and strongest sector of the FoodOmicsGR_RI, bringing together researchers from the mass spectrometry and the NMR spectroscopy fields. FoodOmicsGR_RI researchers have developed methods for metabolic profiling of wine, cheese, honey, royal jelly, olive oil, olives and several other foods, natural products and biological samples.

Untargeted methods, combined with fit-for-purpose chemometric approaches, strengthen the breadth of traditional targeted analysis [25,26] and reveal new prospects for food authenticity investigations. Combining universal analytical approaches with high throughput analytical techniques such as High-Resolution Mass Spectrometry (HRMS), we were able to develop feasible untargeted methodologies for the large-scale determination of unknown compounds in food fingerprinting studies [9,27,28,29,30]. The coupling of Liquid Chromatography with time-of-flight (TOF) mass spectrometry has proved its powerful analytical performance in these studies, offering a satisfying combination of selectivity and sensitivity at high resolution and subsecond scan speeds [28]. However, without involving advanced chemometrics, the number of detected m/z features may be chaotic, preventing the unambiguous discrimination between food samples. The use of chemometrics enables detailed and comprehensive metabolic chemical profiling of food. The most meaningful m/z features are introduced to the developed workflow for further statistical evaluation. Thus, complete exploitation is achieved, enabling discrimination and classification of food sample populations, and prediction of class memberships using supervised and unsupervised prediction models [30]. Cutting-edge methodologies and workflows were developed and applied in suspect and non-target screening approaches. Among those in-house developed tools, the most significant were “RetTrAMS”, a Quantitative Structure-Retention Relationship (QSRR) prediction model, predicting the retention time of new compounds [31,32] and the “TrendTrAMS” software [33,34], enabling the automated detection of contaminants that present a trend of detection across the tested samples (trend analysis) [35]. Fast and more accurate identification of unknowns can also be performed via the “AutoSuspect” workflow, including sample screening with a wide-scope regulatory database of more than 40,000 chemicals, automated subtraction of analytical procedural blanks, isotopic fitting measurement and modified MS/MS similarity score [31,33].

The validation of findings and the development of Quality Control (QC) measures have been a major element of our research. FoodOmicsGR_RI laboratories have been central in the development of QC approaches and measures for LC-MS based metabolomics [36,37,38,39,40,41,42]. Untargeted metabolomics of foods or complex biological fluids such as blood-derived samples (plasma or serum), urine, or tissue extracts, remains challenging for the researchers. In such analytical efforts, it should first be demonstrated that the generated data are of high quality. The application of such measures, thorough data scrutiny and strong documentation, assures that the data provide real, trustworthy and useful differentiations or patterns among samples and detected features. This assurance can provide the base to proceed with the data for detailed multivariate statistical analysis that would identify potential biomarkers. Straightforward and pragmatic “quality control (QC)” procedures have been extensively studied to allow investigators to monitor the analytical processes employed for global, untargeted, metabolic profiling. The FoodOmicsGR_RI consortium has a strong interest and invests substantial research effort in the validation of findings, the development and the application of stringent quality control criteria and the harmonisation of methods and protocols, as a means to guarantee the validity of findings. In this aspect, FoodOmicsGR_RI staff have extensively studied the effect of the various sample preparation methods on the metabolic profile, examining the extraction of different specimens [40,43,44,45,46,47,48].

As the metabolomics field is maturing, there is increased interest in the development of targeted methods [49]. Ultimately, analyses aim at the goal of quantitative results that could be compared among batches, runs and laboratories. R&D activities in areas such as control of authenticity and geographic origin, development of advanced food products, and quality and health claims of foods/dietary patterns, require solid, unambiguous data on the actual concentration of bioactive compounds. Our consortium has invested significantly in the development and the validation of targeted methods that employ GC-MS/MS or LC-MS/MS analysis and provide quantitative results for a selected list of biomolecules. Such lists may be specific to key metabolite groups such as aminoacids [50], saccharides, organic acids or bile acids [51], or of wider coverage that typically determine close to 120 metabolites from various metabolite groups [52]. Application of these protocols includes studies of royal jelly [53,54], wine [55,56,57], flour, carobs and other foods, as also seen in Section 3.5, as well as biological samples from nutritional studies (as seen in Section 3.6). Targeted methods are thoroughly validated in the specific specimen of interest, applying strict criteria for method validation and quality control. The field was recently reviewed by our group [49].

#### 3.3.2. NMR Based-Metabolomics

NMR spectroscopy is a method of choice for the analysis of complex mixtures as it permits simultaneous monitoring of all constituents within the inherent sensitivity limits. Numerous applications have been proposed in food classification and origin determination in combination with multivariate statistical analysis techniques.

We have used multinuclear and multidimensional NMR spectroscopy as a tool for food profiling, quality control and analysis of a variety of food products. In olive oil, the initial focus was the profiling of low molecular weight compounds that can be used as molecular markers of olive oil quality and provided information on olive oil age, degradation and storage history [58,59]. Based on ^1^H and ^31^P NMR spectroscopy, metabolomics approaches that deal with the authentication of extra-virgin olive oil against adulteration with seed oils, olive oils of low quality were developed and validated [60,61]. Currently, NMR methodologies for the quantification of bioactive phenolic compounds in high quality extra virgin olive oils as well as the examination of correlations between phenolic profiling and the organoleptic properties of olive oil and edible olives are under development. Moreover, sophisticated workflows and statistical methods were developed and applied, mainly in non-target screening approaches, for identification and dereplication of chemical markers and biomarkers. Among these in-house tools, STOCSY (Statistical Total Correlation SpectroscopY) deserves to be mentioned [62]. Successfully applied in human biological fluids, this methodology can significantly assist the metabolomic analysis in foods.

Wine analysis is another field of great interest, and we have developed and applied analysis protocols for metabolite profiling of red and white wines. For wines produced from grapes of different Cretan vine varieties on consecutive wine production seasons, we were able to classify commercial wines based on variety [63]. We have also studied the effect of barrel type on wine aging and the organoleptic properties of wine based on both metabolite and phenolic profiling changes during aging for up to one year [10]. Along similar lines, NMR-based metabolomic models have been developed aiming to assess the geographic authentication of Cretan graviera cheese, and the chemical composition changes during extended aging of graviera cheese. Current work is focused on the detailed metabolite profiling of a series of different cheeses produced locally, and the use of NMR for the determination of adulteration using dried milk powder for cheese production.

NMR employing 1D and 2D experiments remains the most powerful tool in unambiguous structures elucidation of small molecules [64]. Combined with the isolation and purification facilities of our infrastructure, NMR has been widely employed for the characterisation of marker compounds [65] such as secoiridoids and related compounds from olive olive oil, crocins from saffron [66], γ-oryzanols from rice bran oil [67], sesamin and sesamolin from sesame oil [68] or phytonutrients from wild edible greens [69].

Through the acquisition of new NMR instrumentation by FoodomicsGR, the consortium has now access to a novel solid-state NMR spectrometer that offers the capability of NMR analysis of solid foods in their native state without the need for long and arduous extraction procedures. We are already working on developing suitable solid-state NMR analytical protocols, with our initial targets being foods such as cheese, seeds, olive fruits and meat, where the potential for a strong analytical impact in quality control and novel metabolomics method development is very promising.

#### 3.3.3. Genetic and Genomic Analyses for Traceability of Animal Organisms

FoodOmicsGR_RI provides Genomic services through a Genetic traceability system that is based on the identification of animals and their products through the study of DNA. DNA is detectable in every cell, resistant to heat treatments, and allows for individual or species identification. The DNA molecule has the feature of being enormously variable among individuals, making it possible to distinguish among them [70,71]. Once the DNA is extracted from biological material (it can either be animal tissue, blood, muscle, or even a processed food such as canned products or meat products), it is analyzed by molecular markers to obtain a fingerprint or specific allelic frequencies, allowing for individual, breed or species identification. Many different markers have been discovered and studied; at present, the most widely used are single nucleotide polymorphism (SNP) [72]. As already mentioned, DNA analysis provides different levels of identification: the individual one is of great interest and it is strictly linked to food safety, while breed and species discrimination are interesting to detect fraud [71] or certify PDO and PGI products (see below).

At the species level, our research team has developed a “DNA barcoding” method for the genetic identification of fish species in order to detect phenomena of involuntary or deliberate mislabeling of fish products. Specifically, DNA sequencing and universal primers are used for the species determination in different fish products (fresh, frozen, processed, etc.) of various fish species (e.g., salmon, sardine, sea bream, sea bass, trout, etc.). Reliable species identification is accomplished through comparison with big databases such as the BOLD Database, providing 100% accuracy in most cases [73]. For the development of this method, we followed all the acceptance criteria (specificity, sensitivity, repeatability and reproducibility) required for ISO 17025:2017 certification. Additionally, we developed a qualitative and quantitative method for species identification in meat products by using PCR and real-time PCR. We follow a species-specific approach in order to detect mislabeling phenomena and the presence of undeclared species. We can identify the most frequently used species in meat products in Greece (e.g., pig, beef, chicken and lamb). The quantitative approach can discriminate the unintentional presence of an undeclared species (due to the use of improperly cleaned equipment) from the intentional.

Furthermore, geographic traceability genetic tools are developed within the Foodomics consortium, accomplished by a large genomic reference database that our research team has built through various research projects for a number of fish species, birds, and large mammals. In particular, for the two most common commercial fishes in Southern Europe, the sea bream and the sea bass, we have developed, in the framework of FP7 project “AquaTrace: the development of tools for tracing and evaluating the genetic impact of fish from aquaculture”, efficient, high fidelity and cost-effective genomic traceability tools, i.e., SNP panels, for the identification of the origin stock (natural/farmed) of random individuals (https://fishreg.jrc.ec.europa.eu/web/aquatrace). For the sea bass, the panel of the 20 most informative SNPs demonstrates assignment accuracy over 90% for natural and over 60% for farmed individuals [74]. For the sea bream, the accuracy is even higher, with the panel of 15 SNPs performing with a mean assignment accuracy of 93.5% for natural individuals and 83.7% for farmed ones [75].

Genomic analyses are additionally applied to livestock species (e.g., sheep and goat), which constitute an important economic sector for our country, for their genetically assisted breeding. The genomic improvement of the species of interest regarding their products, such as milk yield and quality, meat, wool, etc., is widely performed through single nucleotide polymorphisms (SNPs), which scan the whole genome to identify regions associated with the desirable phenotypic characters. In the framework of iSAGE Horizon 2020 project: “Innovation for Sustainable Sheep and Goat Production in Europe” (https://www.isage.eu/), we have analyzed the genome of the Chios Greek sheep breed, which is well known for its high milk yield characteristics as well as its fertility. Specifically, genome-wide association analyses (GWAS) are performed to identify specific genomic regions on the sheep genome associated with novel derived phenotypes regarding the future selection of animal producers unaffected by weather fluctuations due to climate change. In particular, genetic and genomic characterization of performance resilience to climate volatility under heat stress conditions will be presented regarding Chios dairy sheep. Indicatively, genetic correlations between performance resilience and lifetime milk production and also identification of additive SNPs spanning a region of fifth chromosome and associated with Chios resilience will be reported [76]. Similar genome-wide association analyses will be conducted in the near future regarding other traits of the Chios breed, such as prolificacy, functional longevity and lamb survival, which contribute to evaluating the overall animal performance in order to carry out the genetic improvement in a multi-trait context. Additionally, a total of 1300 sheep and goat samples consisting of seven Greek breeds have already been included in the genome-wide genotyping process in order to perform genomic analyses regarding their adaptation to local environmental conditions. The above-mentioned study is being held as part of the framework of the SMARTER Horizon 2020 project: “Small Ruminants breeding for Efficiency and Resilience” (https://www.smarterproject.eu/).

#### 3.3.4. Proteomics Analysis

Proteomic Analysis within the FoodOmicsGR_RI consortium is focused on dairy products and, in particular, cheese. In this aspect, state-of-the-art omic technologies are extensively used, as previously described [19,20]. Specifically, high-resolution-mass spectrometry (HRMS) nano-LC MS/MS Orbitrap Elite is used for the proteomic and peptidomic analysis of Greek dairy products including local cheeses such as feta, graviera, etc. Proteome and peptidome are very important information for the characterization of the uniqueness of Greek cheese [77]. The analysis is targeted at the detailed identification of the proteome and peptidome content of the major Greek cheeses. For the complete characterization of these products, the study aims to include the proteomic and peptidomic analysis of the milk used for their production and a serial sample analysis during production and maturation. Preliminary results indicated that the quality and the uniqueness are related to the milk used [78].

Cheese has a diverse and complicated microbial community, so understanding the composition of this community and its impact on the quality and safety of cheese products is crucial for the geographical origin determination. Cheese samples are highly heterogeneous and, in some cases, their matrices may be incompatible with the analytical methods. Microbial analysis is a powerful tool for the evaluation of food safety and quality that may be used through the whole agri-food chain. The reduction in foodborne hazards is another big concern (i.e., potentially pathogenic microorganisms, pathogens and spoilers) for the improvement of food quality, the prevention of food spoilage and the extension of shelf-life by limiting microbial interactions [3,79]. Furthermore, the microbial traits of the milk used for manufacturing the cheeses are factors that can influence the quality of the final product [21,80].

#### 3.3.5. Elemental Metabolomics for Food Authentication

Elemental Metabolomics have been developed in collaboration with the University of Boston and Griffiths University [81]. Elemental metabolomics is emerging as an important new technology with applications in medical diagnosis, prognosis, nutrition, agriculture, food science, and a multitude of other areas. Elemental Metabolomics based on Elemental Mass spectrometry [82] has been applied for food authentication [83], as well as determination of geographic, genetic and processing origin. Applications concern the geographical and botanical type of honey [84], authentication of Greek Protected Designation of Origin cheeses [85], modulation of egg metallome with flavonoids [86], authentication, safety and nutrition of Graviera Cheese [87], and determination of meat production methods (wild vs. farmed) [88]. Elemental metabolomics applied to PDO “Fava Santorinis” [89] revealed that rare earth elements’ minimal harvest year variation facilitates robust geographical origin discrimination, while data fusion with trace elements discriminates “Fava Santorinis” from other yellow split peas [90]. Greek olive oils’ geographic origin characterization has been achieved through rare-earth content [91]

### 3.4. Sample Banking

FoodOmicsGR_RI includes expert application groups and end-users with deep knowledge of specific food products and large, well-controlled historic sample banks of food products produced in a controlled environment with full historic records (e.g., wine, honey, royal jelly, olive oil, dairy products, Chios mastic, mountain tea, etc.). All manipulations were performed by specialised research scientists working under strictly controlled conditions; samples were collected from specific restricted geographical areas, and were treated and stored in the same way.

Sample banks containing honey and royal jelly samples were produced by specialised Apiculture Laboratories in experimental breeds. Olive oil collection has been built through participation in research projects and national initiatives and covers specific geographical areas, e.g., the islands of Crete and Lesvos, and mainland regions. Monocultivar olive oils have been collected after thorough control that ensured universal oil production technology. In a similar way, monocultivar wines have been collected from collaborating wine makers. Regarding animal tissue/DNA samples, our research team has built through various research projects a thorough reference database for several fish species as well as other species like birds and mammals. Partner laboratories maintain various sample banks of non-genetically modified soybeans cereals, nuts, almonds and other foods. Sample collections extend to several years, to enable the generation of representative profiles. Samples are accompanied by metadata: breed, species, exact geographic location of the farm, farming/feeding modes, and so forth.

As an example, the process for the collection of milk and dairy products will be described in brief: Specialist groups (veterinary scientists) organised sample collection from well-known collaborating animal farms from certain geographical regions; detailed reporting of animal feed, animal breed, age was applied along with the application of strict sample collection, transportation and storage protocols. The collection of milk was organised from specific farms in various areas of Greece. For cheese samples, all production steps, from milk sampling until the cheese making, were monitored by trained scientists and veterinarians. Specifically, the hygienic condition of the farms was examined, taking into consideration the animal breeds, the territory of origin (Epirus, Thessaly, Macedonia and Western Greece), the quality of the milk and the seasonal variation. Milk was selected from different flocks, with different grazing and production systems, whereas all zootechnical parameters were recorded. Cheeses were manufactured according to specific recipes, in absolutely strict conditions of safety and quality, with a fixed percentage of ovine and goat milk in the feta and graviera cheeses, the single breeds used, as well as the cheese-making technology, which was also recorded before the experimental analysis. One sample bank consists of 180 different milk samples from 180 different flocks of small ruminants corresponding to six restricted geographical areas and 140 produced cheeses in total, whereas 48 single breed cheeses were produced in certain regions, generating specific profiles to understand the actual differences among breeds, production methodologies and territory influences. Eighty commercial cheese products from four different regions were also compared, based on our strategic plan to differentiate the authentic and traditional way of cheese making from the industrial methods [92,93,94]. In addition, an independent parallel collection of milk samples from Central Macedonia was organised to include more than 100 milk samples from cow, sheep, goat, donkey and buffalo.

### 3.5. Food Analysis, Traceability and Control of Geographical Origin

This research line aims to study the potential of omics technologies in the control of authenticity (authentication) of agricultural products, the identification of characteristic patterns of biomarkers and/or the certification of the geographic origin of foods. An impressive array of analytical techniques has been used in the literature for this quest. Omic technologies (mass spectrometry, next-generation sequencing, NMR, etc.) and bioinformatics are currently being put to use, aiming to achieve unparalleled detailed food analysis, whilst providing useful information regarding the molecular content of foods. 

In the following section, selected examples of the work of the consortium are provided according to the analytical technology.

#### 3.5.1. LC-MS and GC-MS Technology

Our consortium has extensively applied MS-based metabolic profiling to categorise foods such as wine, olive oil, olives, milk, apples grapes, Chios mastic and other foods. Authenticity studies regarding adulteration and geographical and varietal origin have been efficiently addressed by applying different MS-based screening strategies (target, suspect and non-target) [9,27,28,29,30,95,96]. Hence, efficient classification and discrimination methodologies were developed regarding food samples of different variety, geographical/animal origin or agronomic background, by combining analytical information with advanced chemometrics [28,29]. Integrated MS-based workflows are continuously being developed by our consortium in food matrices of high nutritional value, such as olive oil, wine, dairy products, honey, etc. For example, apple variety classification is proposed based on volatiles analysis by HS-SPME-GC-MS [97]; grapes were classified according to their varieties using HILIC-MS/MS targeted mode [48]

One of the most challenging tasks in food research is labeling verification, mainly regarding products characterized by PDO/PGI indication. An indicative example is the discrimination of PDO Kalamon olive drupes from drupes of different geographical origin, based on their full metabolome [30]. Moreover, reliable methodologies of high sensitivity have already been developed, enabling the detection of potential food adulteration [9]. Taking full advantage of MS-high applicability in the field, significant authenticity markers have been detected and identified through targeted and untargeted approaches, achieving complete characterization of the chemical profile for each target food sample.

The proteome domain can also be studied and used for the assessment of geographical origin. In a preliminary study on feta cheese samples from different areas of Greece, the full protein content of feta cheese was reported for the first time. A total of 489 proteins of diverse functions were detected and were linked to the identity of the product [98]

#### 3.5.2. NMR Technology

Consortium members have developed several analytical protocols based on ^1^H and ^13^C NMR spectroscopy for metabolite profiling of a series of different food products [63,99,100,101,102,103,104]. A special protocol based on ^31^P NMR spectroscopy after chemical derivatization was developed and was used to profile minor glyceride components in olive oil [58,59], and was subsequently applied successfully for the analysis of a variety of foods [105]. Using this protocol for NMR metabolite profiling, multivariate statistical analysis models were developed for the detection of adulteration of extra-virgin olive oil with different seed oils [60] or olive oils of inferior quality, such as lampante and refined olive oils. Furthermore, this NMR-based metabolomics approach was used for the geographical characterization of cv. Koroneiki virgin olive oils produced in different Greek regions [106].

For wine variety and region classification, FoodOmicsGR_RI members developed protocols for the analysis of the phenolic content after extraction using XAD technology resins [107]

#### 3.5.3. MALDI-TOF-MS Technology

Cheese samples (feta, kefalograviera, graviera and goat cheese) were investigated by the FoodOmcsGR_RI staff for the presence of a specific type of bacteria, which can determine the quality and sensory characteristics, and they were also authenticated in four different geographical areas of Greece. Conventional approaches used to identify different types of bacteria include microbiological techniques based on the morphological, biochemical, physiological (phenotype) and genetic characteristics (genotype) of the microorganisms [18,108]. Identification of all isolates using MALDI-TOF MS provides the bacterial “fingerprint”, which allows for accurate identification of bacterial presence. The main proteins used for identification are ribosomal proteins along with some housekeeping proteins. Acquired bacterial MS fingerprints are matched against spectral libraries previously collected under identical MALDI conditions without further identification. For the correct identification of species, a generated peak list was matched against the established reference library, using the integrated pattern-matching algorithm of MALDI Biotyper software [109,110,111]. Appendix A, provide examples of bacterial species with different effects on the final product that concern the quality and safety of the tested cheese samples. Appendix A gives a Phyloproteomic tree (dendrogram) of Lactobacillus rhamnosus and Lactobacillus paracasei, revealing the bacterial diversity according to the milk origin. This technology can also be employed to study the microbial diversity of PDO cheeses to establish which components are responsible for the authentic taste, flavors, and textures of these products.

#### 3.5.4. Elemental Metabolomics Technology

Applications of Elemental Metabolomics within our consortium include: determination of geographical and botanical type of honey [84], authentication of Greek Protected Designation of Origin cheeses [85], modulation of egg metallome with flavonoids [86], authentication, safety and nutrition of Graviera Cheese [87], and determination of meat production methods (wild vs. farmed), [88]. Elemental metabolomics applied to PDO “Fava Santorinis” [89] revealed that rare earth elements’ minimal harvest year variation facilitates robust geographical origin discrimination, while data fusion with trace elements discriminates “Fava Santorinis” from other yellow split peas [90]. Greek olive oils’ geographic origin characterization has been achieved through rare earths content [91]

#### 3.5.5. Data Combination

The work of the consortium aims to combine metabolomics data from different platforms NMR, LC-MS, GC-MS to comprehensively map the metabolome of the foods under study. A further plan is to combine data across omics: for example, metabolomics with elemental analysis (ICP-MS) data, genetics and proteomics analysis data, and so on. Fusion of the data will be used to provide an integrated perspective of the studied specimen. FoodOmicsGR_RI staff have coined a novel term for such holistic approaches for food analysis: “Trophometry”, that is, the sum of omics techniques encompassing analysis of foods. The “Trophometric Fingerprint” is expected to be unique for any given food sample, and can provide information regarding traceability of raw materials as well as the origin and quality of each food itself [112].

In addition, given the ability for sensory evaluation of food products in available premises, omics disciplines will assist in the elucidation of sensory-active compounds, considering all possible stimuli of multimodal perception (aroma, taste, texture, etc.) and including the biological perception of food as an additional analytical dimension to comprehensively assess the quality of a given product.

Figure 2 provides a view of the planned combined analyses and data fusion, aiming to attain comprehensive and holistic characterization.

### 3.6. Nutritional Studies, Highlighting the Value of Greek Foods

In nutrition Research and Development (R&D), our strategy is to exploit the power of Omics technologies to assess domestic agri-food biodiversity for the consolidation of the value of Greek produce [106]. Based on the findings obtained from action described in Section 3.1 (Database of Greek Food Constituents) and action Mapping the composition of Greek foods and Greek Products (described in the previous Section 3.5), we aim to propose, assist and guide biotechnological interventions to highlight the value of Greek Foods. This can be realised by studying the beneficial effects of food consumption to the consumer, monitoring the effect of food ingredient(s) at the genomic–transcriptomic–proteomic–metabolic–elemental level, validating findings and finally developing proof of concept studies on food constituent bioavailability or bioactivity and its effect on human health.

This research line will analyse biological samples from animal models or humans after nutritional intervention. Within the facility, long expertise exists in profiling biological fluids and excreta such as urine, blood, feces, saliva [37,113,114,115] and tissue samples (liver, kidney, brain and others) [42,116]. Profiling of biological samples has been extensively used to provide evidence for nutritional intervention studies that include the application of diet on human and animal models [115,116,117,118,119].

Increased interest in foodomics is central in life/medical science towards the prevention of disease, the development of a personalised diet and the development of nutraceuticals and functional foods. In this line of research, FoodOmicsGR_RI partners realise collaborative projects with various partners from the private sector to improve the manifestation of quality claims of existing products or support R&D efforts toward the development of novel products containing bioactive ingredients from the Greek flora/marine organisms (natural colours, antioxidants, sweeteners or probiotics that help the control of gut microbiota). In a recent example, functional foods containing phenolic-rich water extracts of olives generated during olive oil production, or plant extracts, were prepared, and their effect on animal and human metabolic models was evaluated [120]. For foods of animal origin, the work includes R&D actions guided by Omics techniques that identify new targets for exploitation as a feed or study disease biomarkers in livestock biological samples (e.g., Project Fitsow on sow health, GSRT 2019–2021). The consortium staff have access to the new state of the art animal facility of the Aristotle University that includes staff with expertise on the design of nutritional studies on humans.

Members of the FoodOmicsGR_RI consortium have investigated the health benefits of different constituents of emblematic Greek traditional foods like olive tree polyphenols for their cardioprotective, anti-ischemic and hypolipidemic effects [121,122,123,124,125,126,127] as well as their activity in combination with well-known chemotherapeutics [121,122,123,124,125,126,127,128]. The characterisation of foods’ constituents, quality or chemical markers and dereplication of known compounds are amongst our research axis and applications. So, we have also elaborated on the identification of key metabolites in different types of chocolate, mountain tea decoctions, saffron, Chios mastic gum, as well as on the very characteristic in Mediteranean diet edible greens (chorta) such as “stamnagathi” [66,69,99,100,101,102,103].

### 3.7. Valorization of Agro-Industry by-Products

The agro-industry produces important amounts of solid and liquid by-products that are characterized, on the one hand, by high organic loading, while, on the other, by the presence of bioactive compounds. The future sustainable management of these by-products should focus on the application of management practices that achieve protection of the environment and simultaneous increases in revenues. Under this frame, the amounts of the agro-industrial wastewater that are produced in Greece will be estimated, while different physicochemical [129] and biological processes using microalgae and macrophytes [130] will be applied for the recovery of bioactive compounds from wastewater and the production of valuable biomass.

FoodOmicsGR_RI members have developed value chains in the food-agricultural industry that promote recycling economy by taking into account regional needs. These value chains are based on the integration of innovative technologies for recycling and extracting fine chemicals of high added value from constituents of agricultural wastes. The recovery of natural products with valuable health benefits like anti-oxidants, etc., can be achieved from agricultural wastes at an industrial scale using state-of-the-art green technologies like XAD adsorption resins or supercritical fluid extraction and used in numerous applications in food supplements, cosmeceuticals, etc. These applications can create not only new products from the food and agricultural industry but also lead to significant reductions in environmental footprint, enabling sustainable agriculture practices. Several applications have been developed for olive-oil mills, wineries and wood treatment. [131]

Additionally, FoodOmicsGR_RI partners are highly equipped for the treatment of agriculture wastes and bioproducts in three [131,132,133,134,135,136,137] different scales, i.e., lab-, large- and pilot-scale [131,132,133,134,135,136,137] for the recovery of high added value compounds. The results of this process lead to the procurement of total extracts as well as isolated compounds that can be further evaluated for their properties and exploited from pharma, agro and cosmetic companies for the development of new or optimised products [138]. Indicative examples include the valorisation of olive mill waste waters and edible olives, debittering water for the isolation of polyphenol extracts, the γ-oryzanols from rice bran [67], vinification by-products from wineries, pistachio pericarps, [131], grape pomace polyphenols, antimicrobial properties of Chios mastic water by-product. One of the best-documented cases is on the strong cardioprotective effect of hydroxytyrosol, which is the major constituent of edible olive debittering water.

### 3.8. Assessment of Food Safety

FoodOmicsGR_RI focuses also on the assessment of food safety from chemical contaminants which can be present in foodstuffs due to environmental contamination, cultivation practices or production, packaging and storage processes. One major task deals with the development, application and validation of sensitive, accurate and robust analytical methods based on various extraction techniques (such as solid-phase extraction, Quechers, sonication, etc.) and hyphenated quantitative techniques (such as LC(GC)-MS/MS and LC(GC)-HR-MS/MS) using state-of-the-art instrumentation for facing current and future needs of food surveillance and for identifying new aspects to be controlled. Different classes of traditional and emerging contaminants such as pesticides, biocides, pharmaceuticals, PAHs, phthalates, organohalogenated compounds, etc., as well as their metabolites and transformation products, are studied in terms of their fate, dissipation, levels in food commodities, compliance with current maximum residue limits (MRLs) and possible associated dietary risks. A greater interest is provided for the most significant food matrices of the Greek production market such as olive oil, wine, honey, milk, vegetables, aquaculture products, etc. [139,140,141,142,143,144,145].

In food safety assessment, strong interest also exists in the study of the effect of molecules (additives, oligomers or others) migrating from food contact materials (FCM) to food or food simulants. As such extensive work has focused on the determination of endocrine disruptors, different approaches have been studied including migration of the potential endocrine disruptor, bisphenol A (BPA), from polycarbonate baby bottles into aqueous food simulants [146], as well as the simultaneous determination of butyl- and octyltin compounds in two aqueous-based food simulants [147]. Additionally, different analytical approaches have been studied for the investigation of food migration including simultaneous determination of specific metals in canned tomato paste before and after opening [148], and evaluation of quality indicators for the estimation of the shelf life of opened cans using the migration of specific metals as variables [149].

Regulated substances and non-intentionally added substances (NIAS) migrating into food or official food simulants have been studied by different analytical approaches covering BPA, phthalates [150] and other NIAS [151]. 

It should be mentioned that in the framework of national and industrial food safety control, a wide range of the developed methodologies has been accredited by the Hellenic Accreditation System (E.SY.D) according to ISO 17025 and has been established for the in-depth monitoring of contaminants in different food matrices [152,153,154,155,156] as well as for the analysis of pharmaceuticals and other substances in biological samples [157]

## 4. Impact on the National Research Environment

The establishment of the RI offers benefits for the Greek research system in conducting cutting edge research in Foodomics research topics. FoodOmicsGR_RI can provide a reference center, a center of excellence for the analysis of the molecular/elemental content of food products, with its spearhead on small molecule analysis (metabolomics). The consortium can assist and strengthen research proposals and R&D efforts of the RI partners, external researchers, private enterprises and stakeholders.

Very important issues in food quality, namely fraud and authenticity, are recognised as major activities in different types of food, while the health benefits of specific nutrients attract the interest of a wide research community, as well as producers, retailers, and consumers. In this aspect, FoodomicsGR_RI can assist in the integration of major areas that are important for the development of new research in different areas: botanical-agricultural, veterinary, analytical, nutritional biological effects and food engineering and related technologies. The variety of methodologies covered within the RI permits the establishment of testing cascades for high-throughput information-rich screening methods suited for juridical proof of fraud, authenticity and quality control. This direction can offer an efficient, cost-effective strategy for surveillance and consumer protection, making it possible to harmonize future standards and guidelines for their implementation. This can attract collaboration from scientists from the regulatory bodies in the region, thus increasing the impact on the Greek society and national economy.

## 5. Concluding Remarks

The recent establishment of FoodOmicsGR_RI promotes agri-food and nutrition R&D in Greece, bringing the national research system in line with large European and global initiatives. The activities of FoodOmicsGR_RI promote effective networking and collaboration among the consortium partners, but also between the research and the private sector. The unique combination of facilities, controlled sample banks, databases and expertise is available to support researchers and entrepreneurs in the realisation of R&D efforts.

Omics research necessitates collaborative work from different disciplines and standpoints. Apart from the infrastructure, and the portfolio of analytical methods, this activity required specialists’ knowledge and high-quality sample banks, and FoodOmicsGR_RI combines these characteristics.

FoodOmicsGR_RI addresses a priority for Greece because the country possesses unique biodiversity, as described above in detail. The characterization of unique traditional products, highlighting their value and the potential benefit of the Greek/Mediterranean diet, is best achieved by an evidence-based analytical description of their compositional characteristics with the application of advanced technology. This can increase the attractiveness and market value/demand for these products, thus supporting brand naming and generating higher revenue for the producers and national economy.

## Figures and Tables

**Figure 1 metabolites-11-00074-f001:**
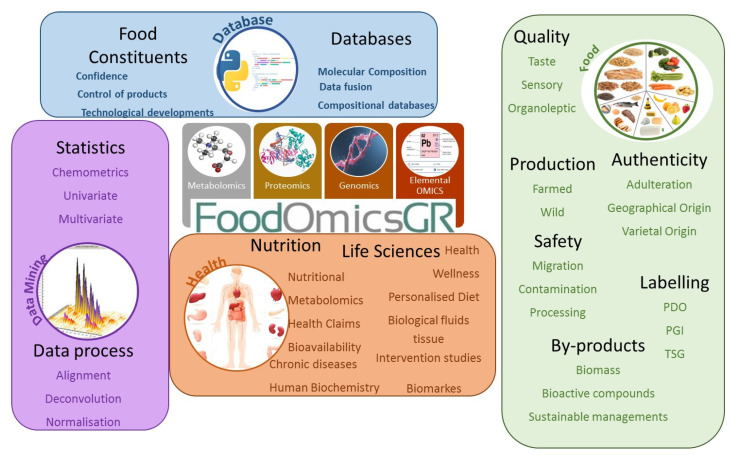
Schematic illustration of the major technologies of the FoodOmicsGR_RI in the center, where the data are generated (metabolomics, genomics, proteomics, elemental metabolomics), and the different action points and work-packages on the periphery where the data are further treated by statistics and then organized in databases. At the right, the application in Food characterization toward authenticity, quality evaluation safety and waste re-use. At the bottom, the studies that aim to highlight the nutritional value of Greek foods. In black/bold, the priority areas; in coloured fonts, areas with potential for synergy and development.

**Figure 2 metabolites-11-00074-f002:**
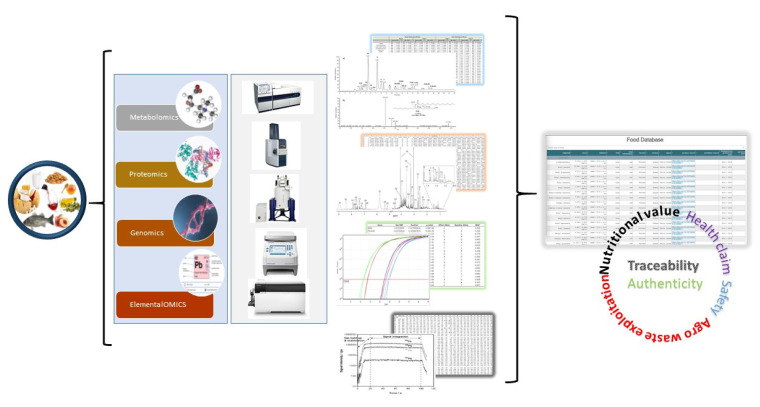
Illustration of the sample path. FoodOmicsGR_RI focuses on the multi-omics analysis of samples by an array of modes in four omics fields (genomics, proteomics, metabolomics and elemental metabolomics) and the fusion and combination of the data to create a comprehensive atlas of the food content for selected Greek foods.

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
