# Peer review of "FoodOmicsGR_RI: A Consortium for Comprehensive Molecular Characterisation of Food Products"

_metabolites, 2021, doi:10.3390/metabo11020074_

Round 1
Reviewer 1 Report
The manuscript shows an extensive and detailed description of the consorptium FoodOmicsGR_RI. The consorptium has an incredible compilance of powerful techniques in all multi-omics approaches, making a holistic tool for any kind of food analysis, which can be also implemented in other research lines. The manuscript depicts the potential of multidisciplinary teams. However, in some parts of the manuscript I've found some English errors.
I include a list of them:
1-Introduction:
l.1.: "IN" the recent years
l.7: new and novel are synonims. Please remove one of them
Second paragraph, despite having the ideas well defined, have really long sentences (one of them has 7 lines). I encourage authors to reduce sentences in this paragraph.
- FoodOmicsGR_RI plan and strategy
Second paragraph: Can the authors mean “It is an ambition that Omic approaches have been offered to…”?
Page 6:
When naming Journal Metabolites, at the end of the sentence it is repeated. Place MDPI in the beginning of the sentence and put out Metabolites MDPI in the end of it or vicecersa.
2nd Paragraph before 3.2: “while it can also reduce”
3.2. Advancing Analytical Capability of the RI.
There is 2 paragraphs by error. It should only be one.
Figure 1: Metabolomics is the unique word without capital letter, in the orange rectangle.
3.2.2. NMR based-metabolomics
1st line: “for the analysis OF complex…”
2nd paragraph, 5th line: “deal with”
2nd paragraph, 7th line: This sentence is difficult to understand. I encourage to change “development and the examination” by “development as well as the examination”.
Page 10, penultimate paragraph, line 3: As you are citing published results, change “NMR is widely” by “NMR have been widely”.
3.3.3. Genetic and Genomic analyses for traceability of animal organisms
2nd paragraph, line 5: At the end of the enumeration, etc is without the letter c.
Page 13, 3.5.1. LC-MS and GC-MS technology, 2nd paragraph, 2nd line: insert a space between by and PDO/PGI.
Page 13, 3.5.1. LC-MS and GC-MS technology, 3rd paragraph: In the second sentence a verb is missing.
3.5.1. Last paragraph: I encourage authors to mix the last paragraph with 3.3.4 subsection, as the work is quite similar al looks a bit repetitive.
3.5.4. 1st sentence: A verb is missing.
3.5.5. 1st paragraph, last line: origin is repeated.
In addition, I've found different writting style qualitites in the whole manuscript, and I think that it should be homogeneized before publication.
Reviewer 2 Report
The review article by Theodoridis et al. introduces the consortium “FoodOmicsGR_RI” constituted by 8 different Greek research institutions whose main aim is provide society and agri-food producers with a set of technologies to characterize the composition and nutritional value of their products. This manuscript is the description of what the consortium is able to provide and what are its goals. A summary of findings by the different research groups from the consortium is included for the different areas covered, which shows a broad and comprehensive overview. I can recommend publication after addressing my comments. See below:
I somehow miss the background as why this first started, are there any similar initiatives in Greek? Or elsewhere? This would be interesting for readers.
I found some subjective and redundant expressions. Some examples are:
-Page 3. “highest biodiversity in Europe”: it would help to support this by literature. When searching for this online, Spain or Italy are the first hits.
-It is not necessary to state that the research is conducted by experienced researchers or that researchers in charge of the experiments have a specific expertise, this is already implied in this type of studies and thus it may appear as redundant. Examples of this are pages 8, 12, 15, among others.
Some minor details:
I would suggest to be consistent on the used terminology. For example, R&D is used but also separated by spaces, i.e. R & D. Then, on page 15 R&D is again described.
I found some typos in the Supplementary data . In the structure of the Facility it reads recongised instead of recognized (AUTh paragraph). On UoC paragraph, do authors mean endogenous? It reads indigenous products?
Supplementary tables show the available protocols within the consortium. Since this is a review article I would suggest that authors include published literature as examples for the described protocols.
During the main body of the manuscripts authors sometimes refer to “paragraphs” for instance on page 5, it is stated “(paragraph 3.5)”, do authors mean section?
Page 6 refers to an in-house mining algorithm for literature filtering, is this publicly available?
Page 10. PDO and PGI abbreviations were already described earlier (on page 3). Same on page 14.
Author Response
"Please see the attachment."

Round 2
Reviewer 1 Report
The whole manuscript has been homogeneized and now it's ready for publication.